# Non-viral derivation of induced pluripotent stem cells from the canine umbilical cord

Atsushi Yamazaki[1], Seiji Shiozawa [1,2], Yuko Koushige[1], Hirotaka Kondo[3], Hisashi Shibuya[3], Kazuya Edamura [1]*

1 Laboratory of Veterinary Surgery, Department of Veterinary Medicine, College of Bioresource and Sciences, Nihon University, Fujisawa, Kanagawa, Japan, 2 Institute for Disease Modeling, Kurume University School of Medicine, Kurume, Fukuoka, Japan, 3 Laboratory of Veterinary Pathology, College of Bioresource Sciences, Nihon University, Fujisawa, Kanagawa, Japan

* edamura.kazuya@nihon-u.ac.jp

## Abstract

In our previous study, canine induced pluripotent stem cells (iPSCs) were successfully generated from skin-derived fibroblasts, without the use of viral vectors. However, for clinical application of canine iPSCs in veterinary regenerative medicine, iPSCs generated from less invasive cell sources would be desirable. Therefore, the purpose of this study was to generate iPSCs from canine umbilical cords discarded at fetal birth. Canine umbilical cords were cut into small pieces and cultured in Dulbecco's modified Eagle's medium supplemented with 10% fetal bovine serum. Episomal vectors carrying 10 reprogramming gene sets were introduced into fibroblasts obtained from the umbilical cord using electroporation. When putative iPSCs colonies emerged, constitutive cell characterization was performed to evaluate cell morphology, proliferative potential, alkaline phosphatase staining, expression of stem cell markers, and the ability to differentiate into a trilineage following embryoid body and teratoma formation. Multiple putative iPSC colonies formed when reprogramming gene sets were introduced into fibroblasts obtained from the canine umbilical cord. The resulting colony cells stained positive for alkaline phosphatase, and showed expression of OCT4, SOX2, NANOG, SSEA1, and SSEA3 on fluorescent immunostaining for stem cell markers. Furthermore, the mRNA expression of canine endogenous *OCT4*, *SOX2*, and *NANOG* significantly increased, confirming the multi-differentiation potential of cells after embryoid and teratoma formation. In this study, iPSCs were successfully generated from canine umbilical cord without the use of a viral vector. Furthermore, canine umbilical cord-derived iPSCs were successfully cultured in a feeder-free manner. This study contributes to the development of veterinary regenerative medicine by using canine iPSCs.

**Data availability statement:** All relevant data are within the manuscript and its Supporting Information files.

**Funding:** This work was supported by JSPS KAKENHI Grant Numbers JP22H02524 and JP23756806.

**Competing interests:** The authors have declared that no competing interests exist.

## Introduction

Induced pluripotent stem cells (iPSCs) have the potential to infinitely self-renew and differentiate into all somatic cell lineages. Human iPSCs were first generated by Takahashi et al. in 2007 [1] using a retroviral vector, while virus-free iPSCs were subsequently established to allow clinical applications [2]. Furthermore, a method for culturing human iPSCs without using feeder cells or animal-derived components has more recently been established [3], while many clinical studies using such iPSCs are currently underway [4–7]. Recently, researchers have recognized that human iPSCs need to be generated from less invasive cell sources to allow their clinical application. Therefore, iPSC generation from less invasive cords and peripheral blood has become mainstream [8], with these cells being used in clinical trials [9].

Several studies had previously reported on the establishment of canine iPSCs using viral vectors [10,11]; however, virus-free canine iPSCs were not established. Therefore, the clinical-grade canine iPSCs generation method using episomal vectors was established, and virus-free canine iPSCs were successfully generated in 2021 [12,13]. Furthermore, it was demonstrated that feeder-free and chemically-defined culture systems could stably maintain canine iPSC [14]. As such, the generation of clinical-grade canine iPSCs will further accelerate research into their clinical applications in veterinary medicine.

To apply these generated canine iPSCs in veterinary medicine, it is necessary to prepare many cell lines and select those that are most likely to differentiate into cells suitable for treatment. However, skin-derived fibroblasts are the most common source of cells to generate canine iPSCs, meaning that skin tissue must be frequently harvested from dogs to generate many iPSC lines. Frequent skin sampling of dogs is undesirable from an animal welfare perspective, as it causes considerable stress. The ideal source of canine iPSCs for use in veterinary medicine is somatic cells, which can be harvested without invasive procedures to facilitate the generation of iPSCs. Therefore, this study focused on the canine umbilical cord as a potential source. As umbilical cords are usually discarded at birth, juvenile cells obtained from umbilical cords are considered suitable for the generation of canine iPSCs. In the present study, canine umbilical cords were isolated and cultured, while canine iPSCs were generated by introducing episomal vectors into umbilical cord-derived cells.

## Materials and methods

### Ethics statement

The experiment of in vivo three-germ-layer differentiation potential of canine iPSCs through teratoma formation was performed in accordance with the guidelines for laboratory animals set forth by the National Institute of Health and the Ministry of Education, Culture, Sports, Science, and Technology (MEXT) of Japan, and was approved by the Kurume University Animal Experiment Committee (approval number: 2024-096).

## Culture of canine umbilical cord-derived fibroblasts

Canine umbilical cords and placentas, generally discarded at birth, were obtained from a breeding facility (NICHIIGA-KKAN Co., Ltd.). The collected tissues were immediately placed in PBS (FUJIFILM Wako Pure Chemical Corporation, Osaka, Japan) supplemented with 1% Penicillin/Streptomycin solution (P/S; FUJIFILM Wako Pure Chemical Corporation) for storage at 4°C. All subsequent tissue processing procedures were performed on a clean bench.

The placenta and umbilical cord were placed in a 10 cm dish and washed with PBS. The blood vessels present in the umbilical cord were isolated, and the tissue was cut into small pieces for seeding into 10 cm dishes (Fig 1). The tissues were cultured with 'M10 medium', containing Dulbecco's Modified Eagle Medium (DMEM; Thermo Fisher Scientific Inc., MA, U.S.A.) supplemented with 10% inactivated fetal bovine serum (FBS; Thermo Fisher Scientific Inc.) and 1% P/S, in a $CO_2$ humidified incubator at 37°C, 5% $CO_2$. When the umbilical cord-derived fibroblasts proliferated, they were detached and cryopreserved as follows: After discarding the medium, fibroblasts were washed with PBS once and incubated with 0.25% Trypsin EDTA (FUJIFILM Wako Pure Chemical Corporation) at 37°C for 3 min. The M10 medium was then added to a 10 cm dish, after which cells were collected in 15 ml tubes, and centrifuged at $200 \times g$ for 3 min. The supernatant was discarded, cells were suspended in STEM CELL BNAKER (NIPPON ZENYAKU KOGYO Co., Ltd., Fukushima, Japan), and cryopreserved at −80°C using BICELL (Nihon Freezer Co., Ltd., Tokyo, Japan). The next day, the cryopreserved cells were transferred to a liquid nitrogen tank and stored until use for the generation of iPSCs.

## Generation of canine iPSCs from umbilical cord-derived fibroblasts

Umbilical cord-derived fibroblasts were expanded on a 0.1% gelatin (NACALAI TESQUE, Inc., Kyoto, Japan)-coated tissue culture dish in M10 medium supplemented with 10 ng/ml recombinant human basic fibroblast growth factor (bFGF; NACALAI TESQUE, Inc.).

Nucleofector 2b (Lonza, Basel, Switzerland) and the Basic Fibroblast Nucleofector Kit (Lonza) were used for vector transfection, as previously described [12]. In brief, a total of 3.65 µg DNA vectors including 0.47 µg pCE-hOCT3/4 (Addgene, MA, U.S.A, #41813), 0.47 µg pCE-hSK (Addgene, #41814), 0.47 µg pCE-hUL (Addgene, #41855), 0.47 µg pCE-mp53DD (Addgene, #41856), 0.47 µg pCXLE- EGFP (Addgene, #27082), 0.47 µg pCE-K2N (Addgene, #154879), 0.47 µg pCE-KdGI (Addgene. #154880) and 0.36 µg pCXWB-EBNA1 (Addgene, #37624) were introduced into $1 \times 10^6$ fibroblasts under U-23 condition of the device.

Following pre-expansion of the transfected fibroblasts into M10 medium supplemented with 10 ng/ml bFGF for three days, fibroblasts were detached using 0.25% trypsin-EDTA solution and plated onto mouse embryonic fibroblasts (MEFs; REPROCELL Inc., Kanagawa, Japan) cultured on a 0.1% gelatin-coated tissue culture 6-well plate, at a density

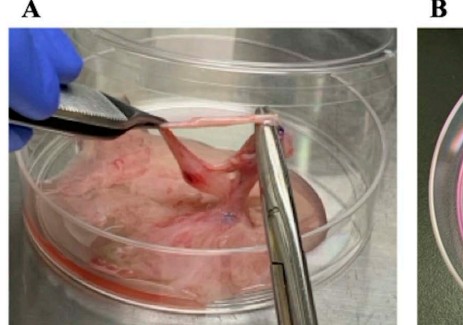
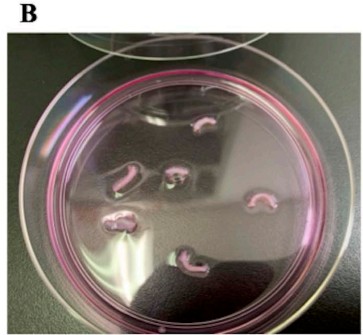

**Fig 1. Isolation and culture of vascular tissue from the canine umbilical cord.** (A) The blood vessels from the canine umbilical cord were isolated with sterilized forceps and scissors. **(B)** The collected blood vessels were then cut into small pieces and seeded into 10 cm dishes.

of $1.5 \times 10^5$ cells per well. Twenty-four hours later, the medium was replaced with 50% M10 medium and 50% naïve ES medium (nESM). After two days, the medium was again replaced with 100% nESM. The medium was then changed every other day until colony selection. The nESM was composed of a 1:1 mixture of Neurobasal (Thermo Fisher Scientific Inc.) and DMEM/F-12 (Thermo Fisher Scientific Inc.), supplemented with 5% Knockout Serum Replacement (KSR; Thermo Fisher Scientific Inc.), 1% N2 supplement (Thermo Fisher Scientific Inc.), 2% B27 supplement (Thermo Fisher Scientific Inc.), 1 mM L-glutamine (L-glu; NACALAI TESQUE, INC.), 0.1 mM MEM Non-Essential Amino Acids Solution (NEAA; NACALAI TESQUE, INC.), 0.1 mM β-mercaptoethanol (2ME; Thermo Fisher Scientific Inc.), 50 µg/ml AlbuMax I (Thermo Fisher Scientific Inc.), 10 ng/ml LIF (NACALAI TESQUE, INC.), 3 µM CHIR99021 (FUJIFILM Wako Pure Chemical Corporation), 1 µM PD0325901 (FUJIFILM Wako Pure Chemical Corporation), 10 µM Forskolin (Merck KGaA, Darmstadt, Germany), and 5 µM A83-01 (FUJIFILM Wako Pure Chemical Corporation). Following three weeks of induction, primary colony-forming cells were mechanically picked and transferred onto new MEFs for further expansion using the Primed ES medium (pESM), supplemented with 10 µM Y-27632 (FUJIFILM Wako Pure Chemical Corporation). The pESM consisted of 1x Knockout Dulbecco's modified Eagle's medium (Thermo Fisher Scientific Inc.), supplemented with 20% KSR, 0.1 mM NEAA, 1 mM L-glu, 0.2 mM 2ME, 1% P/S, and 10 ng/ml bFGF. After three weeks, putative iPSC colonies were mechanically picked and transferred to new MEFs in a 6-well plate. Passaging of iPSCs was performed as follows: confluent iPSCs were detached using KBM Trypsin AOF (KOHJIN BIO CO., LTD., Saitama, Japan), and plated onto new MEFs on a 6-well plate using pESM supplemented with 10 µM Y-27632. The medium was then changed every day.

The feeder-free culture of canine iPSCs was performed as follows: KBM Trypsin AOF was added to iPSCs cultured in MEF-coated 6-well plates and incubated at 37°C for 5 min. iPSCs were collected into a 15 ml tube and centrifuged at $200 \times g$ for 3 min, after which the supernatant was aspirated. The cells were then suspended in StemFit AK03N (AJINO-MOTO CO., INC., Tokyo, Japan), supplemented with 10 µM Y-27632 and plate them onto an iMatrix 511 silk (Matrixome Inc., Osaka, Japan)-coated well. According to the manufacturer's instructions, iMatrix coating was performed at 0.24 µg/cm$^2$. From the day after passage, the medium was changed daily to StemFit AK03N without Y-27632.

## Alkaline phosphatase staining of canine iPSCs

For alkaline phosphatase staining, iPSCs were fixed in 4% paraformaldehyde (FUJIFILM Wako Pure Chemical Corporation) for 10 min at room temperature, while alkaline phosphatase staining was performed using SIGMAFAST BCIP®/NBT (Merck KGaA) according to the manufacturer's protocol.

## Immunocytochemistry of pluripotency markers in canine iPSCs

For immunocytochemical analysis, canine iPSCs were fixed in 70% ethanol (FUJIFILM Wako Pure Chemical Corporation) or 100% methanol (FUJIFILM Wako Pure Chemical Corporation), permeabilized with 0.2% Tween 20 in PBS (PBS-T; Merck KGaA), and blocked with 10% Goat or Donkey serum (Merck KGaA) in PBS, prior to overnight incubation with the following primary antibodies: anti-OCT4 (1:200, Abcam plc., Cambridge, UK), anti-SOX2 (1:100, Abcam plc.), anti-NANOG (1:100, Abcam plc.), anti-SSEA1 (1:100, Abcam plc.), anti-SSEA3 (1:250, Abcam plc.), anti-SSEA4 (1:250, EMD Millipore), anti-TRA-1-60 (1:200, Abcam plc.), and anti-TRA-1-81 (1:200, Abcam plc.). Cells were washed with 0.2% PBS-T at room temperature and incubated with Alexa Fluor 488- or 555-conjugated secondary antibodies (1:400; Thermo Fisher Scientific). The nuclei were stained with Hoechst 33258 (1:1000; Thermo Fisher Scientific, H3569). Details of the antibodies used are listed in S1 Table. The immunolabeled cells were observed under a fluorescence microscope (BZ-X800, KEYENCE CORPORATION, Osaka, Japan).

## RT-qPCR analysis of canine iPSCs

The mRNA expression levels of pluripotency markers in the generated iPSCs and three germ layer markers in the embryoid body (EB) were evaluated using RT-qPCR. The pluripotency markers evaluated included *OCT4, SOX2,* and

*NANOG*, while the trilineage markers evaluated were *PAX6* (ectoderm), *KDR* (mesoderm), and *SOX17* (endoderm). Total cellular RNA was extracted using the RNeasy® Plus Mini (QIAGEN, Hilden, Germany), in accordance with the manufacturer's instructions. Reverse transcription was performed using PrimeScript RT master mix (TaKaRa Bio Inc., Shiga, Japan) with a T100™ Thermal Cycler (Bio-Rad Laboratories, Inc., CA, USA).

RT-qPCR reactions were performed in triplicate with 1 µl (template: 6.25 ng) of the first-strand cDNA using the canine-specific primer sets (TaKaRa Bio Inc.) and TB Green Premix Ex Taq II (TaKaRa Bio Inc.) in the Thermal Cycler Dice® Real Time System II (TaKaRa Bio Inc.). The canine-specific primers (Table 1) were designed using NCBI Primer–BLAST. Each PCR involved 1 cycle of denaturing at 95 °C for 30 seconds, 40 cycles of denaturing at 95 °C for 5 seconds, and annealing and extension at 60 °C for 30 seconds. The results were analyzed using the crossing point and comparative cycle threshold (ΔΔCt) methods in TP900 DiceRealTime v4.02B (TaKaRa Bio Inc.). RT-qPCR of no-template controls was performed with 1 µl Nuclease- and DNA-free water, while amplification of GAPDH from the same amount of cDNA was used as an endogenous control. The specificity of the amplified PCR products was verified using melting curve analysis.

### *In vitro* differentiation of canine iPSCs

*In vitro* differentiation through EB formation was performed as previously described [12]. In brief, subconfluent iPSCs were detached using KBM trypsin AOF, and transferred to floating cultures using Corning Ultra-Low Attachment 6-well plates (Corning, NY, U.S.A) in M10 medium supplemented with 10 µM Y-27632. The cells were then cultured in floating culture for two weeks and further differentiated in adhesion culture in fibronectin (Corning)-coated 24-well plates for two weeks.

For immunocytochemical analysis, EB-derived differentiated cells were fixed with 100% methanol or 70% ethanol, permeabilized with 0.2% PBS-T, and blocked with 10% Donkey serum in PBS, prior to overnight incubation with the following primary antibodies: anti-βIII-tubulin (1:100, Merck KGaA), anti-SOX17 (1:100, R&D Systems, Inc., MN, U.S.A), and anti-αSMA (1:100, Agilent Technologies, CA, U.S.A). The membranes were then washed with 0.2% PBS-T at room temperature, and incubated with Alexa Fluor 488-conjugated secondary antibodies (1:400; Thermo Fisher Scientific). Nuclei were stained with Hoechst 33258 (1:1000). Details of the antibodies used are listed in S1 Table. All immunolabeled cells were observed under a fluorescence microscope (BZ-X800; KEYENCE CORPORATION).

### Assessment of the *in vivo* three-germ-layer differentiation potential of canine iPSCs through teratoma formation

To assess *in vivo* differentiation via teratoma formation, approximately $5 \times 10^6$ canine iPSCs were injected into the testes of NOD/SCID mice (n = 5). After three months, the mice were sacrificed, and the testes were fixed in 4% PFA, stained with hematoxylin-eosin, and used for pathological analysis.

**Table 1. List of the canine-specific primers for RT-qPCR.**

| Name | Gene symbol | NCBI accession number | Forward primers (5′ – 3′) | Reverse primers (5′ – 3′) | Product (bp) |
|------|-------------|----------------------|---------------------------|---------------------------|--------------|
| GAPDH | *GAPDH* | NM_001003142.2 | GATGGGCGTGAACCATGAGA | AGTGGTCATGGATGACTTTGGCTA | 107 |
| OCT4 | *LOC481709* | XM_038682857.1 | CCCTCTGTGTCTGTCACCACTCT | TCTACACCCTTTGTGTTCCCAGA | 184 |
| SOX2 | *SOX2* | XM_038445642.1 | ACAGTTGCAAACGTGGAAAAGAA | AACCTGTATGGCCATTTTTGCTT | 197 |
| NANOG | *NANOG* | XM_038437912.1 | TTCCAGCAAAATTCTATGGGTGA | TAATGGGACACTATCGAGGCAGA | 253 |
| PAX6 | *PAX6* | NM_001097544.1 | TGAGGAACCAGAGAAGACAGGCT | CAGAGGTGAAGGAGGAAACAGGC | 123 |
| KDR | *KDR* | NM_001048024.1 | TGTACCAGACCATGCTTGACTGC | TCTTTGCCATCCTGCTGAGCATT | 118 |
| SOX17 | *SOX17* | XM_038441118.1 | CGACTCTGTTGTGAACCTCCCTG | CCGTCAGACGTCAGGATAGTTGC | 103 |

## Karyotype analysis

The iPSCs were expanded under feeder-free conditions and passaged into T25 flasks. Then, Q-banding-based karyotype analysis was performed by Chromosome Science Labo. Ltd. (Hokkaido, Japan).

## Statistical analysis

In RT-qPCR analysis, the obtained data were calculated as the mean ± standard deviation. Statistical analyses were performed using GraphPad Prism version 6.0 for Macintosh (GraphPad Software Inc., San Diego, CA, USA). The Mann–Whitney $U$ test was applied to compare the two groups. Statistical significance was set at $p < 0.05$.

# Results

## Culture of canine umbilical cord-derived fibroblasts

The vascular tissue of the umbilical cord was cultured in DMEM supplemented with 10% FBS, resulting in the appearance of spindle-shaped cells after a few days (Fig 2). A few days later, the cells were confirmed to have increased in number, and the vascular tissue was removed and cultured. After confirming that the cells were 80% confluent, they were detached and cryopreserved.

## Generation of canine umbilical cord-derived iPSCs

Dome-shaped colonies formed on day 14 following the introduction of episomal vectors into canine umbilical cord-derived fibroblasts cultured with nESM (Fig 3). The culture was then continued with nESM, and on day 25, colonies were picked and transferred to new MEF-coated plates. When the medium was changed to pESM, putative iPSC colonies began to form (Fig 4). These colonies were passaged in pESM and a stable culture was confirmed, after which a feeder-free culture was attempted. Putative iPSC colonies were detached from the MEFs coated plates, dispersed into single cells, and seeded onto plates coated with iMatrix-511 silk. When cultured using StemFit AK03N, the putative iPSCs could be maintained under feeder-free conditions even after more than 25 passages (Fig 5). Finally, four lines of canine umbilical cord-derived iPSCs were successfully generated (HOA1 #1, #2, #3, and #4), of which HOA1#3 was used for further characterization.

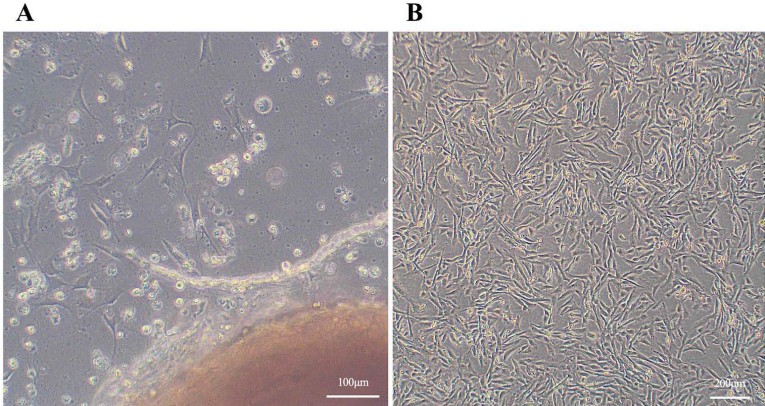

**Fig 2. Culture of canine umbilical cord-derived fibroblasts. (A)** A phase-contrast image of fibroblast outgrowth from the blood vessel tissue of the canine umbilical cord. **(B)** Representative images of proliferated umbilical cord-derived fibroblasts.

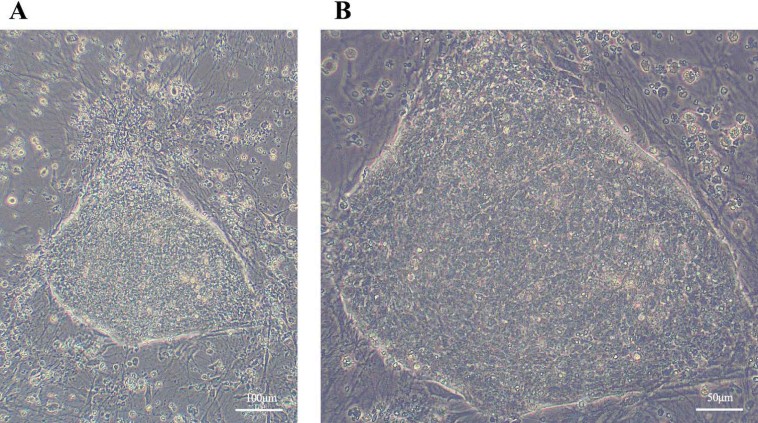

**Fig 3. Primary colonies cultured with nESM.** Representative images of primary colonies generated from canine umbilical cord. **(A)** Low magnification. **(B)** High magnification.

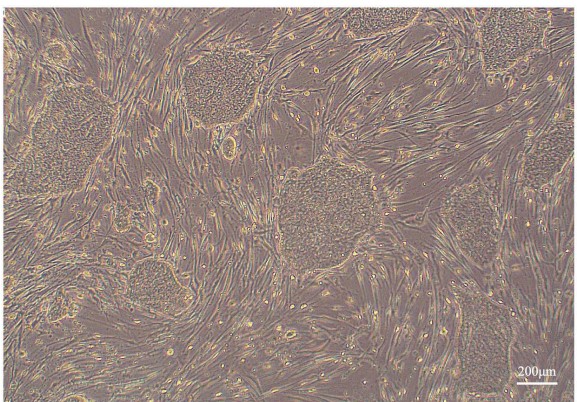

**Fig 4. Putative iPSC colonies cultured with pESM.** Representative images of putative iPSCs colonies generated from canine umbilical cord.

### Expression of pluripotency markers in canine umbilical cord-derived iPSCs

Alkaline phosphatase staining was performed and found to be positive (Fig 5). The expression of pluripotency markers was further confirmed by fluorescence immunostaining, which revealed that OCT4, NANOG, SOX2, SSEA-1, and SSEA-3 were positive, whereas SSEA-4, TRA1-60, and TRA1-81 were negative (Fig 5). The mRNA expression of pluripotency markers in iPSCs was subsequently evaluated, revealing that the mRNA expression of endogenous canine *OCT4, SOX2,* and *NANOG* was significantly higher than that in canine umbilical cord-derived fibroblasts (Fig 6).

### *In vitro* differentiation into three germ layers

When canine iPSCs were cultured in suspension in M10 media supplemented with Y-27632, EBs were formed the next day (Fig 7). EBs cultured in suspension for two weeks were seeded on fibronectin-coated plates and cultured with M10 for another two weeks. Subsequently, the expression of ectoderm (βIII-tubulin), mesoderm (αSMA), and endoderm (SOX17) markers was confirmed by fluorescent immunostaining, and all markers were found to be positive (Fig 7). Following three weeks of floating culture, total RNA was extracted from the EBs. The mRNA expression of ectoderm (*PAX6*), mesoderm

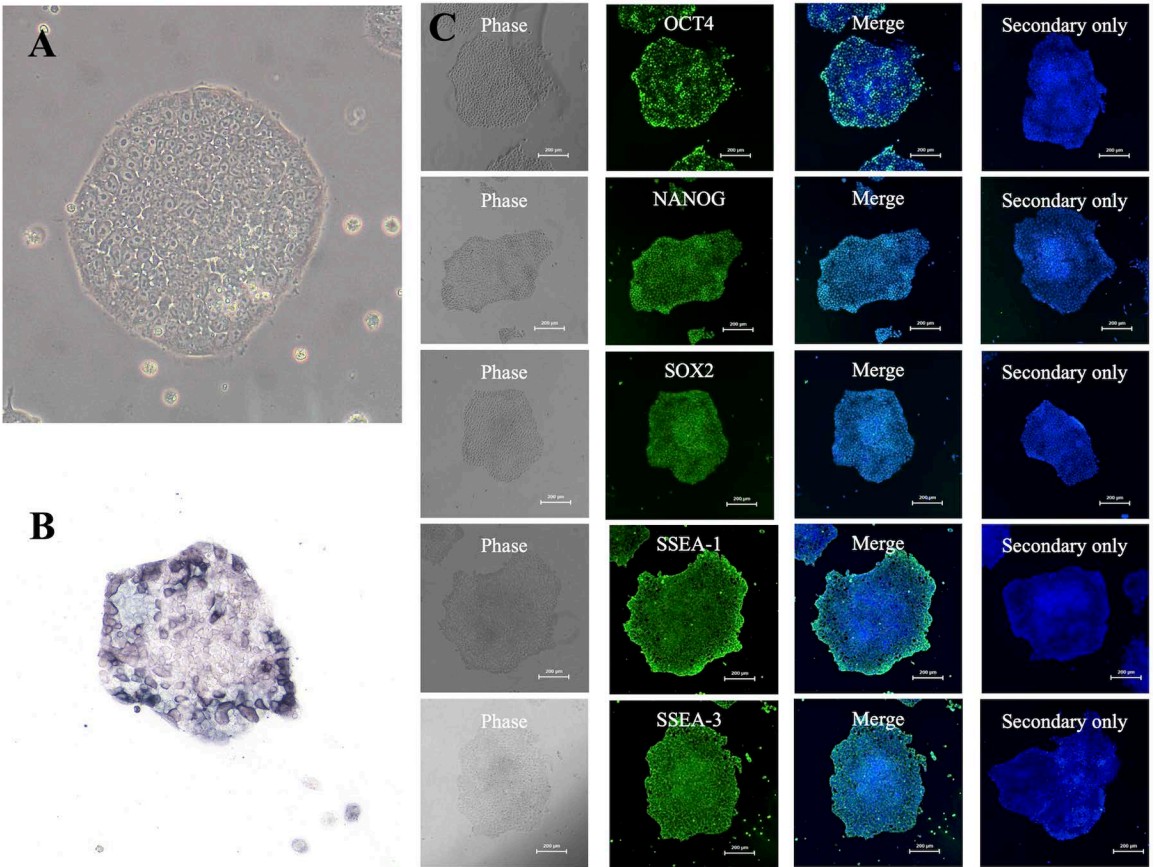

**Fig 5. Canine iPSC colonies growing on iMatrix-511 silk and results of immunocytochemistry of pluripotency markers. (A)** Representative images of canine umbilical cord-derived iPSC colonies on iMatrix. **(B)** Alkaline phosphatase staining of the iPSC colonies was positive. **(C)** Fluorescent immunostaining of pluripotency markers in iPSC demonstrated that OCT4, NANOG, SOX2, SSEA-1, and SSEA-3 were positive.

(*KDR*), and endoderm (*SOX17*) markers was confirmed by RT-qPCR, and all markers tended to increase compared to those in iPSCs (Fig 8). In contrast, expression of the stem cell marker *NANOG* decreased (Fig 8).

### Assessment of the *in vivo* three-germ-layer differentiation potential of canine iPSCs through teratoma formation

To assess the teratoma formation potential of canine iPSCs, these cells injected into the testes of NOD/SCID mice. All mice developed tumors in the testes after 3 months. The excised tumors contained two germ layers, the ectoderm and mesoderm, confirming they were teratomas (Fig 9).

### Karyotype analysis

Q-banding-based karyotype analyses revealed that 10% of the canine umbilical cord-derived iPSCs (5 out of 50 cells analyzed at metaphase) had the normal chromosome number (2n = 78).

### Discussion

In this study, vascular tissue was isolated from the canine umbilical cord, tissue fragments were cultured, and canine umbilical cord-derived fibroblasts were successfully cultured. Canine umbilical cord-derived iPSCs were also successfully

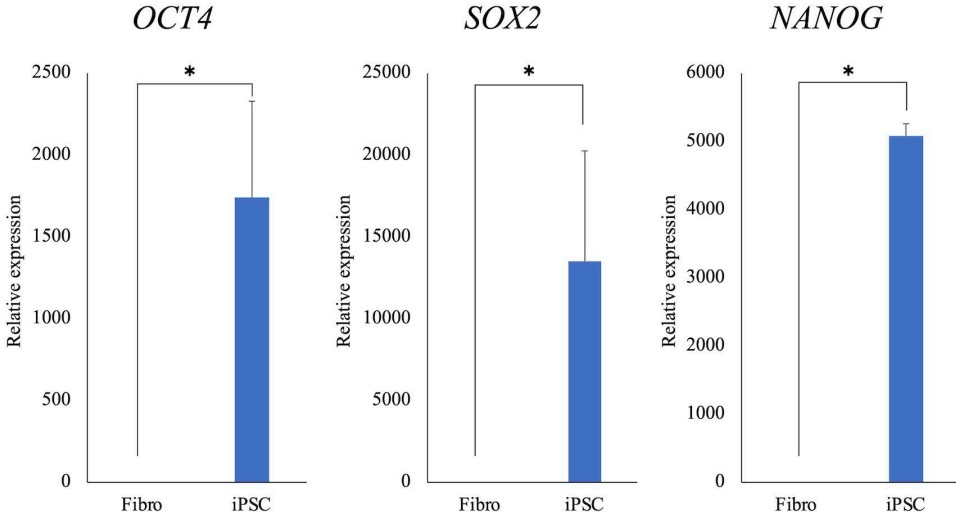

**Fig 6. The mRNA expression of canine endogenous pluripotency markers.** The expressions of canine endogenous OCT4, SOX2, and NANOG mRNA were higher in iPSCs than in umbilical cord-derived fibroblasts. *: Mean values differ significantly between the groups ($p < 0.05$).

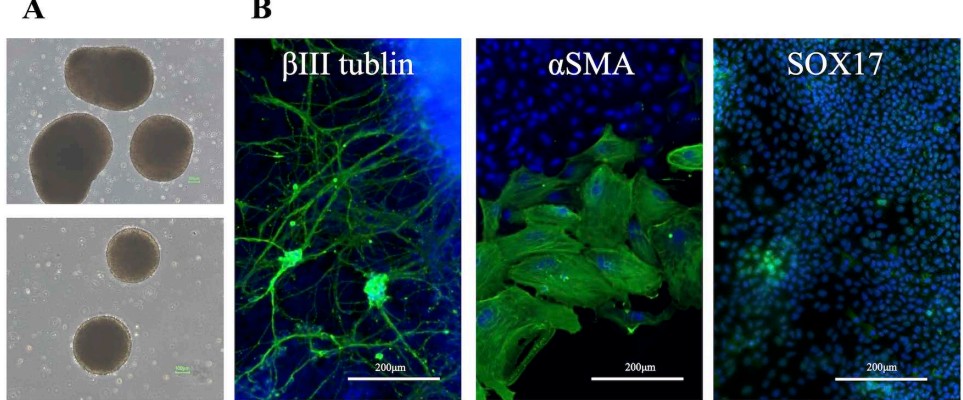

**Fig 7. The morphology and fluorescent immunostaining of trilineage differentiation markers in EBs. (A)** EBs were formed in M10 media the day after seeding iPSCs. **(B)** The expression of ectoderm (βIII-tubulin), mesoderm (αSMA), and endoderm (SOX17) markers were positive.

generated through the introduction of episomal vectors into the cells without viral vectors. This is the first report of the generation of iPSCs from the canine umbilical cord. This study showed that canine iPSCs could be successfully generated from tissues harvested without invasive procedures in dogs. As such, this study is expected to make a significant contribution to the clinical application of canine iPSCs in veterinary medicine.

Unlike the skin tissue, the canine umbilical cord is discarded at birth, and allowing easy harvesting. Therefore, this tissue is a suitable source for the generation of canine iPSCs. In addition, the canine umbilical cord is a fetal appendage, and juvenile cells can be obtained from this tissue. Mesenchymal stromal cells (MSCs) can further be obtained from the canine umbilical cord, while umbilical cord-derived MSCs tend to have high proliferative potential [15]. In addition, it has been reported that MSCs may be included in the fibroblast group as a larger framework [16]. In the present study, canine umbilical cord-derived fibroblasts showed high proliferative potential, similar to that of MSCs. Therefore, it is possible to

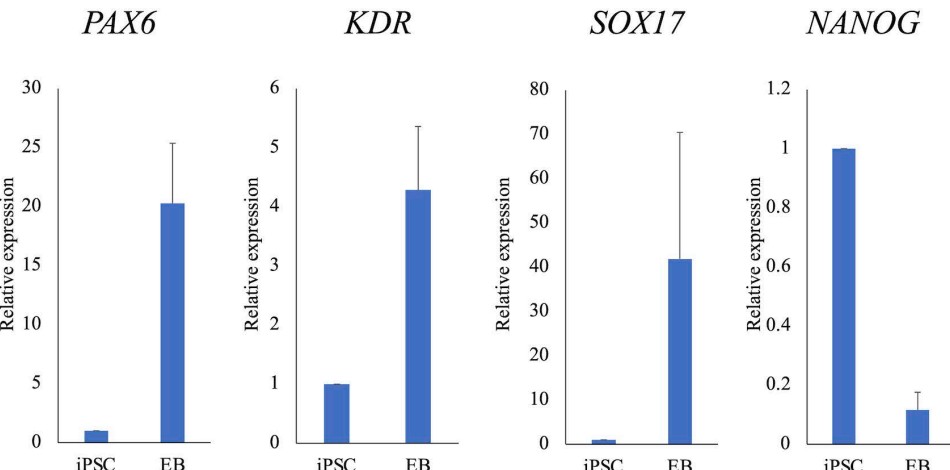

**Fig 8. The mRNA expressions of three germ layers markers.** The expressions of PAX6, KDR, and SOX17 mRNA in EBs tended to be higher than in iPSCs. *: Mean values differ significantly between the groups ($p < 0.05$).

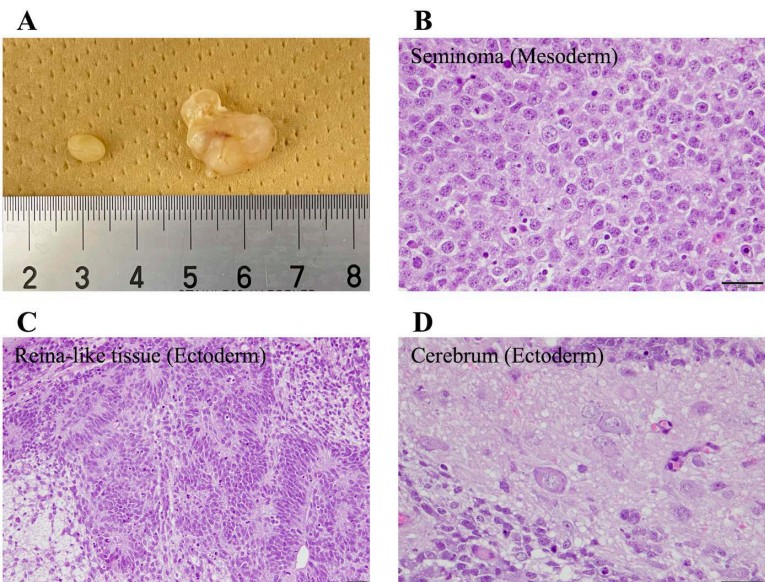

**Fig 9. Teratoma formation ability of canine umbilical cord-derived iPSCs.** Teratomas of the testis were formed in about 3 months. **(A)** Images of the normal testis (left) and testis containing a tumor (right) are shown. **(B)**, **(C)** The teratoma contained a seminoma (mesoderm), retina-like tissue (ectoderm), and cerebrum (ectoderm).

isolate fibroblasts with high proliferative potential from the canine umbilical cord. As the introduction of episomal vectors into cells with high proliferative potential is important for increasing the generation efficiency of canine iPSC, canine umbilical cord tissue may be suitable for the generation of iPSCs.

The colony morphology, proliferative potential, expression of stem cell markers, and multipotency of canine umbilical cord-derived iPSCs were found to be similar to the canine dermal fibroblast-derived iPSCs established in the past [12,13]. In addition, the generation efficiency of canine umbilical cord-derived iPSCs was higher than that of dermal fibroblast-derived iPSCs (data

not shown). As such, this study confirmed that iPSCs could be generated from the canine umbilical cord. Generation of iPSCs from embryonic fibroblasts, skin fibroblasts, and peripheral blood has been reported as a source of canine iPSCs [13,17,18]. One study also showed that canine iPSCs can be generated from peripheral blood by using Sendai virus vectors [18], and their clinical application in veterinary medicine is expected owing to the less invasive cell sources. Furthermore, since this study successfully generated canine iPSCs from the canine umbilical cord using episomal vectors, iPSCs established from less invasive cell sources may be used more frequently in basic and clinical research in veterinary medicine.

The canine umbilical cord-derived iPSCs generated in the present study were cultured using iMatrix-511 silk and StemFit AK03N. Thus, this study is the first to demonstrate that canine iPSCs generated using episomal vectors can be cultured in feeder- and animal-origin-free media. It has further been shown that canine iPSCs generated from peripheral blood using Sendai virus vectors can be cultured using iMatrix-511 and StemFit AK02N [14]. Overall, these experiences show that canine iPSCs can be maintained and cultured in feeder-free and chemically defined media, similar to human iPSCs. Culturing canine iPSCs with the minimal use of animal-derived components is important for their clinical application in veterinary medicine.

When canine umbilical cord-derived iPSCs were analyzed in this study, SSEA1 and SSEA3 were positive, and SSEA4 was negative. Previous studies have reported that human iPSCs are negative for SSEA1, while mouse ES cells are positive [19]. As such, the expression pattern of SSEA1 in canine iPSCs is thought to be similar to that observed in mice. Canine skin-derived iPSCs are negative for SSEA1 [20], whereas canine peripheral blood-derived iPSCs are positive for SSEA1 [18]. Because the cell sources of these canine iPSCs and the vectors used to introduce the reprogramming factors differ, further studies are needed to determine the expression of SSEA1 in canine iPSCs. To the best of our knowledge, there have been no reports on the expression of SSEA3 in canine iPSCs. The SSEA3 expression has been shown to be positive in human iPSCs and negative in mouse ES cells [19], while the present study revealed that the expression pattern of SSEA3 in canine iPSCs is the same as that in human iPSCs. On the other hand, the expression of SSEA4 in canine umbilical cord-derived iPSCs is consistent to previous reports of canine iPSCs [11]. In the present study, canine umbilical cord-derived iPSCs showed flat colony morphology and required bFGF for cell culture, similar to human iPSCs. These results indicate that canine umbilical cord-derived iPSCs may possess the characteristics of human iPSCs; however, further studies are required to determine the exact characteristics of canine iPSCs.

Karyotype analysis of the canine umbilical cord-derived iPSCs showed that 10% of the cells had the normal chromosome number. Because iPSCs with abnormal karyotypes may present a risk of tumorigenesis, it would be desirable to use canine umbilical cord-derived iPSCs with normal karyotypes in veterinary medicine. Although only a few cells had the normal chromosome number in this study, it may be possible to generate canine umbilical cord-derived iPSCs with the normal karyotype by subcloning those cells for clinical application. In addition, the generation of iPSCs from canine umbilical cord tissue of other individuals may lead to the possibility of generating iPSCs with the normal karyotype. Thus, further studies are needed to generate iPSCs with the normal karyotype for the clinical application of canine umbilical cord-derived iPSCs.

One limitation of this study is that further studies are needed to investigate the reproducibility of canine umbilical cord-derived iPSC generation and the characterization of multiple iPSC lines. Canine umbilical cord tissue was used in this study to generate canine iPSCs from tissues that could be harvested without invasive procedures. As such, one important limitation of this study is that it is difficult to use autologous umbilical cord-derived iPSCs unless the umbilical cord was preserved at birth, and allogeneic iPSCs are used for treatment. However, most of the iPSC lines used in human clinical research are allogeneic. Therefore, the canine allogeneic iPSCs generated in this study are expected to have applications in veterinary regenerative medicine.

## Conclusion

In the present study, fibroblasts were successfully cultured from the vascular tissues of the canine umbilical cord. In addition, this study succeeded in generating canine umbilical cord-derived iPSCs by introducing episomal vectors into

the cells. Furthermore, this study demonstrated that canine umbilical cord-derived iPSCs can be cultured in feeder- and animal-free media. Overall, this study presents significant contributions which will promote the development of regenerative medicine using canine iPSCs in veterinary medicine.

## Supporting information

**S1 Table. The primary and secondary antibodies used for immunocytochemistry.**
(PDF)

## Author contributions

**Conceptualization:** Atsushi Yamazaki, Seiji Shiozawa, Kazuya Edamura.

**Data curation:** Atsushi Yamazaki.

**Formal analysis:** Atsushi Yamazaki, Seiji Shiozawa.

**Investigation:** Atsushi Yamazaki, Seiji Shiozawa, Yuko Koushige, Hirotaka Kondo, Hisashi Shibuya.

**Methodology:** Atsushi Yamazaki, Seiji Shiozawa, Kazuya Edamura.

**Project administration:** Kazuya Edamura.

**Resources:** Atsushi Yamazaki, Seiji Shiozawa, Kazuya Edamura.

**Supervision:** Kazuya Edamura.

**Visualization:** Atsushi Yamazaki, Seiji Shiozawa, Kazuya Edamura.

**Writing – original draft:** Atsushi Yamazaki, Seiji Shiozawa.

**Writing – review & editing:** Seiji Shiozawa, Kazuya Edamura.

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
