## [Decision Letter · Decision Letter 0]

PONE-D-24-53708Non-viral derivation of induced pluripotent stem cells from the canine umbilical cordPLOS ONE

Dear Dr. Edamura,

Thank you for submitting your manuscript to PLOS ONE. After careful consideration, we feel that it has merit but does not fully meet PLOS ONE’s publication criteria as it currently stands. Therefore, we invite you to submit a revised version of the manuscript that addresses the points raised during the review process.

We look forward to receiving your revised manuscript.

Kind regards,

Li-Ping Liu

Academic Editor

PLOS ONE

Journal Requirements:

Reviewers' comments:

Reviewer's Responses to Questions

**Comments to the Author**

1. Is the manuscript technically sound, and do the data support the conclusions?

Reviewer #1: Yes

Reviewer #2: Yes

Reviewer #3: Yes

2. Has the statistical analysis been performed appropriately and rigorously? 

Reviewer #1: Yes

Reviewer #2: Yes

Reviewer #3: Yes

3. Have the authors made all data underlying the findings in their manuscript fully available?

Reviewer #1: Yes

Reviewer #2: Yes

Reviewer #3: Yes

4. Is the manuscript presented in an intelligible fashion and written in standard English?

Reviewer #1: Yes

Reviewer #2: Yes

Reviewer #3: Yes

5. Review Comments to the Author

Reviewer #1: The author described successful IPS-derived from umblical cord snalysis by using non-viral vector, and made it for therapeutic condition for transformation likely by using some transcription factors. Author need to be update protein levels likely proteins levels and cell proliferation rate and, also needs to be included chromosomal abnormality even stemness function checked.

therefore, first of all,

1)immunoblot analysis

2)msc differention cell cycle checkpoint

3)chromosomal abnormality

Best,

Reviewer #2: Dear editor,

The MS “Non-viral derivation of induced pluripotent stem cells from the canine umbilical cord” describes the derivation and characterization of canine iPSCs.

The authors provide a thorough iPSC characterization and there is no doubt that iPSC generation was successful. However, one point should be clarified.

For some reason karyotype analysis was not performed. Chromosome stability (or instability) is one of the most important cell type characteristics. Veterinary usage of produced iPSC is possible only if they have normal karyotype. The authors suppose that produced iPSC line in primed. If it has XX karyotype, X-chromosome inactivation analysis would provide important clues to the pluripotency type, naïve or primed. Karyotype analysis must be performed.

I’d suggest to use proofreading services to correct English style.

Overall, the MS needs major revision, as new experimental data are necessary.

Minor points:

L51-53 “however, virus-free canine iPSCs have not yet been established… and virus-free canine iPSCs were successfully generated. [12,13].” There is a logical problem – virus-free iPSCs were not established, but at the same time they are.

L61-61 Actually skin samples could be harvested just once, that would give millions fibroblasts for many iPSC lines. Why multiple harvests?

L313 At the Discussion section shall be a paragraph with the explanation, why there were no endoderm. Is it expected? Is it normal for canine PSC teratomas? I’d expect endoderm derivatives as well as there were several teratomas, not one.

L335-335. Actually, as there are MSCs in the umbilical cord as well as fibroblasts, the iPSCs were possible produced from fibroblasts. Human fibroblasts and MCSs are indistinguishable morphologically and by marker expression, I suppose it applies for canine ones too. The only way to test the cell type is adipo-, osteo- and chondrocyte differentiation.

L386 iPSCs would be autologous to the dog from whose umbilical cord they were produced.

Reviewer #3: In this study, canine iPS cells were generated using episomal vectors from umbilical cord-derived cells. Previous studies have cultured cells in an invasive manner, so the generation of iPS cells from umbilical cord-derived cells, which does not require any invasive procedures, is a very significant study. However, some modifications are necessary.

Major Point,

Canine iPS cells were generated from umbilical cord-derived cells. However, it is unclear whether these are from the umbilical cord of the same individual or from multiple individuals. Generating iPS cells from multiple umbilical cords is desirable to confirm the reproducibility of the experiment.

Page 16, Line 257: "Finally, four lines of canine umbilical cord-derived iPSCs were successfully generated (HOA1 #1, #2, #3, and #4), of which HOA1#3 was used for further characterization." is written, and the characteristics were analyzed from only one cell line. However, it should be investigated whether multiple cell lines meet the properties of iPS cells.

Page 21, Line 343: "In addition, the generation efficiency of canine umbilical cord-derived iPSCs was higher than that of dermal fibroblast-derived iPSCs." is stated, but there is insufficient data on the generation efficiency. Please provide specific data showing that umbilical cord-derived iPSCs have a higher generation efficiency than fibroblasts.

Page 22, Line 363: Data on SSEA-1 and SSEA-3 are shown and discussed. However, data on SSEA-4 is required.

Page 23, Line 379: The limitation states that karyotype staining was not performed. I believe that karyotype staining is necessary in this study as well.

Minor Point

Page 8, Line 114. Is the U-23 condition a device setting?

Page 20, Line 331. In this study, it is stated that fibroblasts were cultured, but are these different cells from MSCs? Please consider whether to prove the cell origin.

Fig. 6. Please explain what the asterisks indicate.

Fig. 8 Was there no significant difference?

6. PLOS authors have the option to publish the peer review history of their article (what does this mean? ). If published, this will include your full peer review and any attached files.

**Do you want your identity to be public for this peer review?** For information about this choice, including consent withdrawal, please see our Privacy Policy .

Reviewer #1: **Yes: ** Dr. Kyu-Shik Jeong

Reviewer #2: No

Reviewer #3: No

---

## [Author Response · Author response to Decision Letter 1]

27 Jan 2025

January 27, 2025

To Editor,

Thank you so much for your insightful review of this study. We agree with all of your comments, and the manuscript was thoroughly revised based on your suggestions. The line number in each response was shown as the number in the revised manuscript.

Reviewer #1

Thank you very much for your careful review of our manuscript. Our responses to your comments are as follow:

The author described successful IPS-derived from umbilical cord analysis by using non-viral vector, and made it for therapeutic condition for transformation likely by using some transcription factors. Author need to be update protein levels likely proteins levels and cell proliferation rate and, also needs to be included chromosomal abnormality even stemness function checked.

therefore, first of all,

1)immunoblot analysis

2)msc differention cell cycle checkpoint

3)chromosomal abnormality

Response:

1)immunoblot analysis

As you suggested, immunoblot analysis may be suitable for measuring protein levels of canine umbilical cord-derived iPSCs. However, the expressions of canine endogenous mRNAs OCT4, SOX2, and NANOG were confirmed by RT-qPCR, and the expressions of stem cell marker proteins were confirmed by fluorescence immunostaining in this study. Therefore, the immunoblot analysis is a future work.

2)msc differention cell cycle checkpoint

Further study is needed to reveal MSC differentiation and cell cycle checkpoint. In addition, these data are being obtained, and the results of additional studies will be published in a new paper.

3)chromosomal abnormality

As you indicated, karyotyping is essential for the clinical application of canine iPSCs. Following this study, experiments were conducted to confirm the reproducibility of iPSC generation. Then, karyotyping of canine umbilical cord-derived iPSCs from other individuals revealed that they contained cell lines with normal chromosome numbers. However, the cell lines in the present study have not undergone karyotyping analysis, which is a topic for future work. Additional revisions are needed to include the results of the karyotype analysis of other cell lines in this paper. 

Reviewer #2

Thank you for reviewing the manuscript and interest in our study. We agree with your comment, and the manuscript was thoroughly revised based on your suggestion.

The MS “Non-viral derivation of induced pluripotent stem cells from the canine umbilical cord” describes the derivation and characterization of canine iPSCs.

The authors provide a thorough iPSC characterization and there is no doubt that iPSC generation was successful. However, one point should be clarified.

For some reason karyotype analysis was not performed. Chromosome stability (or instability) is one of the most important cell type characteristics. Veterinary usage of produced iPSC is possible only if they have normal karyotype. The authors suppose that produced iPSC line in primed. If it has XX karyotype, X-chromosome inactivation analysis would provide important clues to the pluripotency type, naïve or primed. Karyotype analysis must be performed.

I’d suggest to use proofreading services to correct English style.

Overall, the MS needs major revision, as new experimental data are necessary.

Response: As you suggested, karyotyping is essential for the clinical application of canine iPSCs. Following this study, experiments were conducted to confirm the reproducibility of iPSC generation. Then, karyotyping of canine umbilical cord-derived iPSCs from other individuals revealed that they contained cell lines with normal chromosome numbers. However, the cell lines in the present study have not undergone karyotyping analysis, which is a topic for future work. Additional revisions are needed to include the results of the karyotype analysis of other cell lines in this paper.

Minor points:

L51-53 “however, virus-free canine iPSCs have not yet been established… and virus-free canine iPSCs were successfully generated. [12,13].” There is a logical problem – virus-free iPSCs were not established, but at the same time they are.

Response: Thank you for pointing this out. The error in the text has been corrected as follows:

Several studies had previously reported on the establishment of canine iPSCs using viral vectors [10,11]; however, virus-free canine iPSCs were not established. Therefore, the clinical-grade canine iPSCs generation method using episomal vectors was established, and virus-free canine iPSCs were successfully generated in 2021 [12,13]. (line: 50-53)

L61-61 Actually skin samples could be harvested just once, that would give millions fibroblasts for many iPSC lines. Why multiple harvests?

Response: To apply canine iPSCs to treatment, it is necessary to prepare many iPSC lines and select cell lines that can differentiate into cells suitable for treatment. In addition, the properties of iPSCs may vary depending on the individual cell source, and it is possible that no iPSC line suitable for treatment can be obtained from a single dog. Therefore, in order to establish iPSC lines that can be used for treatment, it is necessary to harvest skin tissue frequently from multiple dogs, which is very invasive for the animal. However, canine umbilical cord tissue is waste tissue, which makes it possible to collect tissue from multiple dogs noninvasively. This is a major advantage of using umbilical cord tissue for iPSC generation.

L313 At the Discussion section shall be a paragraph with the explanation, why there were no endoderm. Is it expected? Is it normal for canine PSC teratomas? I’d expect endoderm derivatives as well as there were several teratomas, not one.

Response: In our experience, canine iPSCs tend to have difficulty differentiating into endoderm-derived tissues in teratoma formation. In the past, when we formed teratomas using our established canine skin-derived iPSC lines, endoderm-derived tissues were observed, but the umbilical cord-derived iPSCs in the present study did not differentiate into endoderm-derived tissues. Further investigation is needed to determine whether this is due to differences in iPSC lines or the source tissue of the iPSCs. However, the results of in vitro differentiation into three germ layers were shown in this study. In addition, since differentiation into two or more germ layers is recognized as a teratoma, the iPSCs established in this study were proven to have the ability to form teratomas.

L335-335. Actually, as there are MSCs in the umbilical cord as well as fibroblasts, the iPSCs were possible produced from fibroblasts. Human fibroblasts and MCSs are indistinguishable morphologically and by marker expression, I suppose it applies for canine ones too. The only way to test the cell type is adipo-, osteo- and chondrocyte differentiation.

Response: Recent reports have shown that fibroblasts also can differentiate into bone, cartilage, and adipocytes, making it difficult to clearly distinguish fibroblasts from MSCs even after confirming their ability for differentiation into the three cell types (Soundararajan M, Kannan S. J Cell Physiol. 2018 Dec;233(12):9099-9109.). Since MSCs may be included in the group of fibroblasts as a major framework, we decided to describe them as umbilical cord-derived fibroblasts in this study.

L386 iPSCs would be autologous to the dog from whose umbilical cord they were produced.

Response: Thank you for pointing this out. The error in the text has been corrected as follows:

As such, one important limitation of this study is that it is difficult to use autologous umbilical cord-derived iPSCs unless the umbilical cord were preserved at birth, and allogeneic iPSCs are used for treatment. (line: 401-403) 

Reviewer #3

Thank you for reviewing the manuscript and interest in our study. We agree with your comments, and the manuscript was thoroughly revised based on your suggestions.

Major Point,

Canine iPS cells were generated from umbilical cord-derived cells. However, it is unclear whether these are from the umbilical cord of the same individual or from multiple individuals. Generating iPS cells from multiple umbilical cords is desirable to confirm the reproducibility of the experiment.

Response: As you suggested, it is desirable to confirm reproducibility by collecting canine umbilical cord tissue from multiple individuals and generating iPSCs. Therefore, following this study, we conducted experiments to verify the reproducibility of iPSC generation and succeeded in generating canine umbilical cord-derived iPSCs from other individual samples. In this study, we succeeded in generating the world's first canine umbilical cord-derived iPSCs, and this paper aims to show the properties of the first cell line. Thus, we added to the Discussion that experiments are needed to confirm the reproducibility (line: 391-393).

Page 16, Line 257: "Finally, four lines of canine umbilical cord-derived iPSCs were successfully generated (HOA1 #1, #2, #3, and #4), of which HOA1#3 was used for further characterization." is written, and the characteristics were analyzed from only one cell line. However, it should be investigated whether multiple cell lines meet the properties of iPS cells.

Response: As you pointed out, it is desirable to analyze the properties of multiple canine umbilical cord-derived iPSC lines and present all the data. However, since the purpose of this paper is to show the successful generation of iPSCs from the canine umbilical cord, we presented the data of one iPSC line that could be stably maintained and cultured. Thus, we added to the Discussion that further study is needed to confirm the characteristics of multiple iPSC lines in the future. (line: 391-393).

Page 21, Line 343: "In addition, the generation efficiency of canine umbilical cord-derived iPSCs was higher than that of dermal fibroblast-derived iPSCs." is stated, but there is insufficient data on the generation efficiency. Please provide specific data showing that umbilical cord-derived iPSCs have a higher generation efficiency than fibroblasts.

Response: Thank you for your suggestion. It is necessary to present data showing that the generation efficiency of umbilical cord-derived iPSCs is superior to that of skin. Compared with the generation efficiency data of canine skin-derived iPSCs previously generated in our laboratory, it is clear that the establishment efficiency of canine umbilical cord-derived iPSCs is superior to that of skin-derived iPSCs. However, since the purpose of this paper is to show the successful generation of iPSCs from the canine umbilical cord, the comparison data of the generation efficiency was not included in this study. Therefore, we added “data not shown” to this sentence (line: 350).

Page 22, Line 363: Data on SSEA-1 and SSEA-3 are shown and discussed. However, data on SSEA-4 is required.

Response: As you pointed out, a statement regarding the SSEA4 results is also necessary. In this study, we found that the expression of SSEA4 in canine umbilical cord-derived iPSCs is identical to previous reports of canine iPSCs. Thus, it was reconfirmed that canine iPSCs do not express SSEA4. We added relevant sentences to the Discussion (line: 368-369, 381-383).

Page 23, Line 379: The limitation states that karyotype staining was not performed. I believe that karyotype staining is necessary in this study as well.

Response: As you indicated, karyotyping is essential for the clinical application of canine iPSCs. Following this study, experiments were conducted to confirm the reproducibility of iPSC generation. Then, karyotyping of canine umbilical cord-derived iPSCs from other individuals revealed that they contained cell lines with normal chromosome numbers. However, the cell lines in the present study have not undergone karyotyping analysis, which is a topic for future work. Additional revisions are needed to include the results of the karyotype analysis of other cell lines in this paper.

Minor Point

Page 8, Line 114. Is the U-23 condition a device setting?

Response: We added in the text that U-23 is a device setting (line: 117).

Page 20, Line 331. In this study, it is stated that fibroblasts were cultured, but are these different cells from MSCs? Please consider whether to prove the cell origin.

Response: Recent reports have shown that fibroblasts also can differentiate into bone, cartilage, and adipocytes, making it difficult to clearly distinguish fibroblasts from MSCs even after confirming their ability for differentiation into the three cell types (Soundararajan M, Kannan S. J Cell Physiol. 2018 Dec;233(12):9099-9109.). Since MSCs may be included in the group of fibroblasts as a major framework, we decided to describe them as umbilical cord-derived fibroblasts in this study.

Fig. 6. Please explain what the asterisks indicate.

Response: We added the explain what the asterisks indicate (line:290-291).

Fig. 8 Was there no significant difference?

Response: There was no significant difference. We added the explain what the asterisks indicate (line:311-312).

---

## [Decision Letter · Decision Letter 1]

PONE-D-24-53708R1Non-viral derivation of induced pluripotent stem cells from the canine umbilical cordPLOS ONE

Dear Dr. Edamura,

Thank you for submitting your manuscript to PLOS ONE. After careful consideration, we feel that it has merit but does not fully meet PLOS ONE’s publication criteria as it currently stands. Therefore, we invite you to submit a revised version of the manuscript that addresses the points raised during the review process.

According to the reviewers' comments, the identification of iPSC line characterization is not completed. Especially, the cytogenetical and chromosome analysis of iPSC line should be investigated. 

We look forward to receiving your revised manuscript.

Kind regards,

Li-Ping Liu

Academic Editor

PLOS ONE

Reviewers' comments:

Reviewer's Responses to Questions

**Comments to the Author**

1. If the authors have adequately addressed your comments raised in a previous round of review and you feel that this manuscript is now acceptable for publication, you may indicate that here to bypass the “Comments to the Author” section, enter your conflict of interest statement in the “Confidential to Editor” section, and submit your "Accept" recommendation.

Reviewer #1: All comments have been addressed

Reviewer #2: All comments have been addressed

Reviewer #3: (No Response)

2. Is the manuscript technically sound, and do the data support the conclusions?

Reviewer #1: Yes

Reviewer #2: Yes

Reviewer #3: Yes

3. Has the statistical analysis been performed appropriately and rigorously? 

Reviewer #1: Yes

Reviewer #2: Yes

Reviewer #3: Yes

4. Have the authors made all data underlying the findings in their manuscript fully available?

Reviewer #1: Yes

Reviewer #2: Yes

Reviewer #3: Yes

5. Is the manuscript presented in an intelligible fashion and written in standard English?

Reviewer #1: Yes

Reviewer #2: Yes

Reviewer #3: Yes

6. Review Comments to the Author

Reviewer #1: Authors covers all reviewewr's comments one by one and did work nicley.Therefore his paper is now possible to be publish at this journal. At the same time, author followed by research ethics, publication ethics on right.

Reviewer #2: Dear authors,

You have answered most of my questions, though I disagree with the idea that fibroblasts ans MSCs are the same cell type, I'd suggest at least to mention the articles supporting that notion in the discussion section, as the cell identity of reprogrammed cells is really important for the iPSC generation.

The most important criticism is that you have not provided cytogenetical analysis of the generated iPSCs. Cytogenetics is one of the primary characteristics of the iPSC lines, you must provide it. It is not "a topic for future work" as in your response. And it is not just a technical issue, human and mouse pluripotent stem cells tend to accumulate chromosome anomalies due to certain mechanisms relevant to pluripotent and not somatic cells. Without cytogenetics iPSC line characterization is not complete.

Reviewer #3: The author wrote, "Since the purpose of this paper is to show the successful generation of iPSCs from the canine umbilical cord, we presented the data of one iPSC line that could be stably maintained and cultured. "

If so, the karyotype of this important cell line should be investigated.

If, for some reason, the karyotype cannot be examined in this cell line, please provide the karyotype of another generated cell line.

7. PLOS authors have the option to publish the peer review history of their article (what does this mean? ). If published, this will include your full peer review and any attached files.

**Do you want your identity to be public for this peer review?** For information about this choice, including consent withdrawal, please see our Privacy Policy .

Reviewer #1: **Yes: ** Dr. Kyu-Shik Jeong, Republic of Korea.

Reviewer #2: No

Reviewer #3: No

---

## [Author Response · Author response to Decision Letter 2]

15 May 2025

May 13, 2025

To Editor,

Thank you so much for your insightful review of this study. We agree with all of your comments, and the manuscript was thoroughly revised based on your suggestions. The line number in each response was shown as the number in the revised manuscript. 

Reviewer #1

Thank you very much for your careful review of our manuscript. We have received a lot of input from you, which has been very helpful.

Reviewer #2

Thank you for reviewing the manuscript. The manuscript was revised based on your suggestion.

You have answered most of my questions, though I disagree with the idea that fibroblasts ans MSCs are the same cell type, I'd suggest at least to mention the articles supporting that notion in the discussion section, as the cell identity of reprogrammed cells is really important for the iPSC generation.

Response:

As you suggested, we added a sentence in the Discussion as follows:

In addition, it has been reported that MSCs may be included in the fibroblast group as a larger framework [16]. (line: 346-347)

The most important criticism is that you have not provided cytogenetical analysis of the generated iPSCs. Cytogenetics is one of the primary characteristics of the iPSC lines, you must provide it. It is not "a topic for future work" as in your response. And it is not just a technical issue, human and mouse pluripotent stem cells tend to accumulate chromosome anomalies due to certain mechanisms relevant to pluripotent and not somatic cells. Without cytogenetics iPSC line characterization is not complete.

Response:

As you pointed out, we performed a karyotype analysis and added text to the Materials and Methods, Results, and Discussion sections of the manuscript. (line: 228-231, 327-330, 400-409) 

Reviewer #3

The author wrote, "Since the purpose of this paper is to show the successful generation of iPSCs from the canine umbilical cord, we presented the data of one iPSC line that could be stably maintained and cultured. "

If so, the karyotype of this important cell line should be investigated.

If, for some reason, the karyotype cannot be examined in this cell line, please provide the karyotype of another generated cell line.

Response:

As you pointed out, we performed a karyotype analysis and added text to the Materials and Methods, Results, and Discussion sections of the manuscript. (line: 228-231, 327-330, 400-409)

---

## [Decision Letter · Decision Letter 2]

Non-viral derivation of induced pluripotent stem cells from the canine umbilical cord

PONE-D-24-53708R2

Dear Dr. Edamura,

We’re pleased to inform you that your manuscript has been judged scientifically suitable for publication and will be formally accepted for publication once it meets all outstanding technical requirements.

Kind regards,

Li-Ping Liu

Academic Editor

PLOS ONE

Additional Editor Comments (optional):

Reviewers' comments:

Reviewer's Responses to Questions

**Comments to the Author**

1. If the authors have adequately addressed your comments raised in a previous round of review and you feel that this manuscript is now acceptable for publication, you may indicate that here to bypass the “Comments to the Author” section, enter your conflict of interest statement in the “Confidential to Editor” section, and submit your "Accept" recommendation.

Reviewer #2: All comments have been addressed

Reviewer #3: All comments have been addressed

2. Is the manuscript technically sound, and do the data support the conclusions?

Reviewer #2: Yes

Reviewer #3: (No Response)

3. Has the statistical analysis been performed appropriately and rigorously? 

Reviewer #2: Yes

Reviewer #3: (No Response)

4. Have the authors made all data underlying the findings in their manuscript fully available?

Reviewer #2: Yes

Reviewer #3: (No Response)

5. Is the manuscript presented in an intelligible fashion and written in standard English?

Reviewer #2: Yes

Reviewer #3: (No Response)

6. Review Comments to the Author

Reviewer #2: Dear authors,

The MS was modified and my questions were answered.

The fact that the majority (90%) of iPSC cells have abnormal karyotype would not allow the usage of that iPSC line, but does not change the the data on pluripotency.

Reviewer #3: (No Response)

7. PLOS authors have the option to publish the peer review history of their article (what does this mean? ). If published, this will include your full peer review and any attached files.

**Do you want your identity to be public for this peer review?** For information about this choice, including consent withdrawal, please see our Privacy Policy .

Reviewer #2: No

Reviewer #3: No

---

## [Editor Report · Acceptance letter]

PONE-D-24-53708R2

PLOS ONE

Dear Dr. Edamura,

I'm pleased to inform you that your manuscript has been deemed suitable for publication in PLOS ONE. Congratulations! Your manuscript is now being handed over to our production team.

Kind regards,

on behalf of

Dr. Li-Ping Liu

Academic Editor

PLOS ONE